# Monitoring SARS-CoV-2 Infection Using a Double Reporter-Expressing Virus

Kevin Chiem,[a] Jun-Gyu Park,[a] Desarey Morales Vasquez,[a] Richard K. Plemper,[b] Jordi B. Torrelles,[a] James J. Kobie,[c] Mark R. Walter,[d] Chengjin Ye,[a] Luis Martinez-Sobrido[a]

[a]Texas Biomedical Research Institute, San Antonio, Texas, USA

[b]Center for Translational Antiviral Research, Institute for Biomedical Sciences, Georgia State University, Atlanta, Georgia, USA

[c]Department of Medicine, Division of Infectious Diseases, University of Alabama at Birmingham, Birmingham, Alabama, USA

[d]Department of Microbiology, University of Alabama at Birmingham, Birmingham, Alabama, USA

**ABSTRACT** Severe acute respiratory syndrome coronavirus 2 (SARS-CoV-2) is the highly contagious agent responsible for the coronavirus disease 2019 (COVID-19) pandemic. An essential requirement for understanding SARS-CoV-2 biology and the impact of antiviral therapeutics is a robust method to detect the presence of the virus in infected cells or animal models. Despite the development and successful generation of recombinant (r)SARS-CoV-2-expressing fluorescent or luciferase reporter genes, knowledge acquired from their use in *in vitro* assays and/or in live animals is limited to the properties of the fluorescent or luciferase reporter genes. Herein, for the first time, we engineered a replication-competent rSARS-CoV-2 that expresses both fluorescent (mCherry) and luciferase (Nluc) reporter genes (rSARS-CoV-2/mCherry-Nluc) to overcome limitations associated with the use of a single reporter gene. In cultured cells, rSARS-CoV-2/mCherry-Nluc displayed similar viral fitness as rSARS-CoV-2 expressing single reporter fluorescent and luciferase genes (rSARS-CoV-2/mCherry and rSARS-CoV-2/Nluc, respectively) or wild-type (WT) rSARS-CoV-2, while maintaining comparable expression levels of both reporter genes. *In vivo*, rSARS-CoV-2/mCherry-Nluc has similar pathogenicity in K18 human angiotensin-converting enzyme 2 (hACE2) transgenic mice than rSARS-CoV-2 expressing individual reporter genes or WT rSARS-CoV-2. Importantly, rSARS-CoV-2/mCherry-Nluc facilitates the assessment of viral infection and transmission in golden Syrian hamsters using *in vivo* imaging systems (IVIS). Altogether, this study demonstrates the feasibility of using this novel bioreporter-expressing rSARS-CoV-2 for the study of SARS-CoV-2 *in vitro* and *in vivo*.

**IMPORTANCE** Despite the availability of vaccines and antivirals, the coronavirus disease 2019 (COVID-19) pandemic caused by severe acute respiratory syndrome coronavirus 2 (SARS-CoV-2) continues to ravage health care institutions worldwide. Previously, we generated replication-competent recombinant (r)SARS-CoV-2 expressing fluorescent or luciferase reporter proteins to track viral infection *in vitro* and/or *in vivo*. However, these rSARS-CoV-2 are restricted to express only a single fluorescent or a luciferase reporter gene, limiting or preventing their use in specific *in vitro* assays and/or *in vivo* studies. To overcome this limitation, we have engineered a rSARS-CoV-2 expressing both fluorescent (mCherry) and luciferase (Nluc) genes and demonstrated its feasibility to study the biology of SARS-CoV-2 *in vitro* and/or *in vivo*, including the identification and characterization of neutralizing antibodies and/or antivirals. Using rodent models, we visualized SARS-CoV-2 infection and transmission through *in vivo* imaging systems (IVIS).

**KEYWORDS** coronavirus, fluorescent, infection, luciferase, recombinant, reporter, SARS-CoV-2, transmission

Address correspondence to Chengjin Ye, cye@txbiomed.org, or Luis Martinez-Sobrido, lmartinez@txbiomed.org.

The authors declare a conflict of interest. C.Y. and L.M.-S. are co-inventors on a patent that includes claims related to reverse genetics approaches to generate recombinant SARS-CoV-2.

Severe acute respiratory syndrome coronavirus 2 (SARS-CoV-2) is responsible for the coronavirus disease 2019 (COVID-19) pandemic (1). Since the first reported case in Wuhan, China, SARS-CoV-2 has spread worldwide and has been associated with more than 500 million confirmed cases and over 6 million deaths (https://coronavirus.jhu.edu/map.html) (2), in part due to its innate high transmissibility (3, 4). In the past 2 decades, two other human coronaviruses have been responsible for severe diseases in humans, including severe acute respiratory syndrome coronavirus (SARS-CoV) in 2002 and the Middle East respiratory syndrome coronavirus (MERS-CoV) in 2012 (5, 6). Further, four endemic human coronaviruses are responsible for common cold-like respiratory disease: OC43, NL63, 229E, and HKU1 (7, 8). A unique feature of SARS-CoV-2 compared to known betacoronaviruses is the addition of a furin cleavage site in the viral spike (S) glycoprotein, which is a major contributor to the virus's increased transmissibility and pathogenicity (9, 10). Several prophylactic (vaccines) and therapeutic (antivirals or monoclonal antibodies) options have been approved by the United States (US) Food and Drug Administration (FDA) to prevent or treat, respectively, SARS-CoV-2 infection. These include three vaccines (Spikevax [former Moderna], COMIRNATY [former BioNTech & Pfizer], and Janssen) (11, 12), several therapeutic antiviral drugs (remdesivir, baricitinib, molnupiravir, and nirmatrelvir), and one monoclonal antibody (Mab; bamlanivimab) (13–15). Unfortunately, SARS-CoV-2 has rapidly accumulated mutations, leading to the emergence of variants of concern and variants of interest, jeopardizing the effectiveness of existing preventive and/or treatment options (16–20).

Reverse genetics systems have permitted the generation of recombinant RNA viruses entirely from cloned cDNA, facilitating studies to better understand multiple aspects of the biology of viruses, including, among others, mechanisms of viral infection, pathogenesis, transmission, and disease (21–31). Another application of reverse genetics is the generation of recombinant viruses containing gene mutations and/or deletions that result in viral attenuation for their implementation as safe, immunogenic, and protective live-attenuated vaccines (22, 32–37). Moreover, reverse genetics has been used to generate recombinant viruses expressing reporter proteins, thereby abolishing the need for secondary approaches for viral detection (38–44). In this regard, genetically modified recombinant viruses expressing reporter genes have been generated to monitor viral infection in cultured cells and/or in animal models using reporter expression as a valid surrogate readout for viral infection (24, 38, 45–48). Notably, these reporter-expressing viruses have the potential to be used in high-throughput screening (HTS) settings to identify antivirals or neutralizing antibodies that can inhibit or neutralize, respectively, viral infection and to visualize the dynamics of viral infection in validated animal models using *in vivo* imaging systems (IVIS).

Fluorescent and luciferase proteins are used to generate reporter-expressing viruses and represent ideal choices due to their high sensitivity and stability (49–55). Since these reporter genes have dissimilar characteristics, their selection is largely motivated by the type of study or application. Fluorescent proteins are easily detected when excited by absorbing energy at a particular wavelength, which is subsequently emitted as light at a higher wavelength as the molecules drop to a lesser energy state (54). Hence, reporter viruses expressing fluorescent proteins are typically used for *in vitro* studies to observe cellular localization and/or to identify the presence of infected cells (27, 38, 39, 47, 56). Moreover, fluorescence-expressing recombinant viruses are used to identify the presence of the virus in infected cells in validated animal models using *ex vivo* imaging (38, 39, 45–47). However, *in vivo*, fluorescent signals are often obscured by autofluorescence and have insufficient detection due to light scattering. Conversely, luciferases produce bright and localized signals in live organisms to be monitored in real-time using IVIS and represent a viable surrogate of viral replication (38, 39, 57). Moreover, viruses expressing luciferase genes are more sensitive and convenient for quantitative analyses compared to their fluorescent-expressing counterparts (39, 45, 48). Despite the clear advantages of both fluorescence and luciferase reporter genes, only recombinant viruses expressing either fluorescent or luciferase reporter genes have been previously described in the literature (24, 38, 45, 46, 48). In the past, we overcame this issue by generating dual reporter viruses expressing

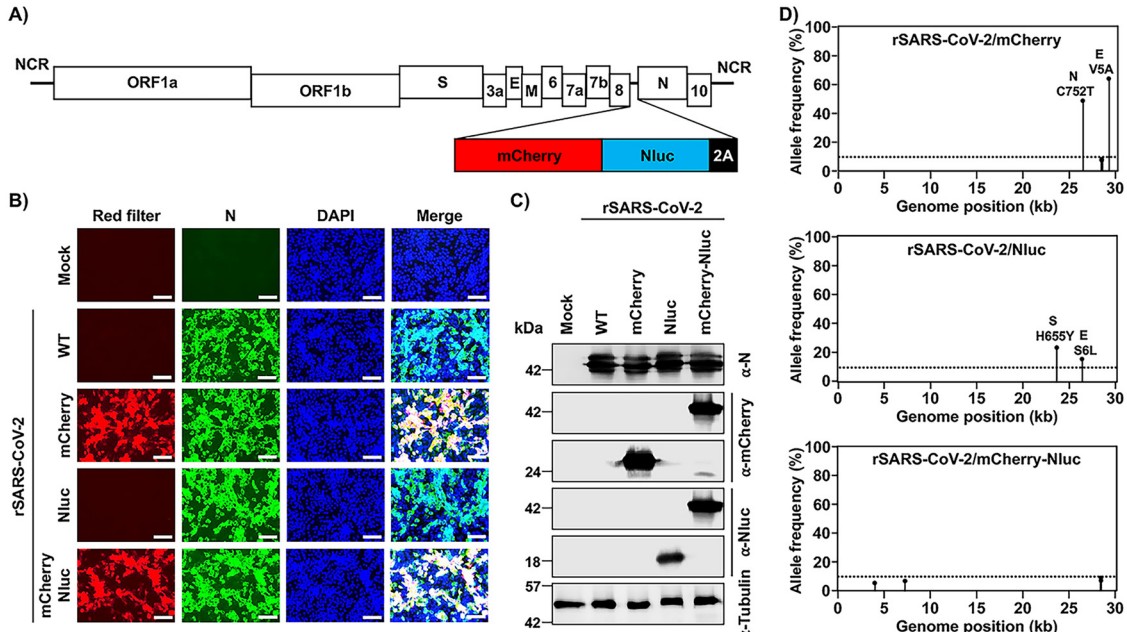

**FIG 1** Generation of a bireporter rSARS-CoV-2 expressing mCherry and Nluc (rSARS-CoV-2/mCherry-Nluc). (A) Schematic representation of the rSARS-CoV-2/mCherry-Nluc viral genome: SARS-CoV-2 structural, nonstructural, and accessory open reading frame (ORF) proteins are indicated in white boxes. mCherry (red), Nluc (blue), and the PTV-1 2A autoproteolytic sequence (black) were inserted in front of the viral N protein. NCR, noncoding region. (B) mCherry expression and immunofluorescence assays: Vero E6 cells ($1.2 \times 10^6$ cells/well, 6-well format, triplicates) were mock infected or infected (MOI, 0.01) with rSARS-CoV-2 WT, rSARS-CoV-2/mCherry, rSARS-CoV-2/Nluc, or rSARS-CoV-2/mCherry-Nluc. Cells were fixed in 10% neutral buffered formalin 24 hpi before directly visualizing mCherry expression under a fluorescence microscope or the viral N protein using a specific 1C7C7 MAb. Cell nuclei were strained with DAPI. Representative images are shown. Scale bars = 100 $\mu$m. Magnification = $\times$20. (C) Western blots: Vero E6 cells ($1.2 \times 10^6$ cells/well, 6-well format, triplicates) were mock infected or infected (MOI, 0.01) with rSARS-CoV-2 WT, rSARS-CoV-2/mCherry, rSARS-CoV-2/Nluc, or rSARS-CoV-2/mCherry-Nluc. At 24 hpi, cells were collected and protein expression in cell lysates were evaluated by Western blotting using specific antibodies against SARS-CoV-2 N protein or the mCherry and Nluc reporter proteins. Tubulin was included as a loading control. The molecular mass of proteins is indicated in kilodaltons (kDa) on the left. (D) Deep sequencing analysis of reporter-expressing rSARS-CoV-2: the nonreference allele frequency of rSARS-CoV-2/mCherry (top), rSARS-CoV-2/Nluc (middle), and rSARS-CoV-2/mCherry-Nluc (bottom) was calculated by comparing the short reads to the respective reference SARS-CoV-2 WA-1 viral genome (MN985325.1). Nonreference alleles present in less than 10% of reads are not shown (dotted line), and the nonreference allele frequency that is greater than 10% is indicated.

both luciferase and fluorescent reporter genes and demonstrated its advantages with influenza and vaccinia viruses (38, 39, 57).

In this study, we used our previously described bacterial artificial chromosome (BAC)-based reverse genetics (21–23, 58, 59) and the innovative 2A approach (45, 46) to pioneer a rSARS-CoV-2 expressing both fluorescence mCherry and luciferase Nanoluciferase (Nluc) reporter genes (rSARS-CoV-2/mCherry-Nluc). Our results demonstrate that rSARS-CoV-2/mCherry-Nluc has similar properties in cultured cells to rSARS-CoV-2 expressing individual mCherry or Nluc reporter genes, or wild-type (WT) rSARS-CoV-2. Importantly expression of the double reporter gene mCherry-Nluc did not affect viral replication or pathogenesis in K18 human angiotensin-converting enzyme 2 (hACE2) transgenic mice or golden Syrian hamsters, validating its use for both *in vitro* and/or *in vivo* studies.

## RESULTS

**Generation of rSARS-CoV-2/mCherry-Nluc.** Recently, we generated rSARS-CoV-2 expressing single reporter genes upstream of the viral N gene using a PTV-1 2A autoproteolytic peptide approach (25). This new rSARS-CoV-2 displayed higher levels of reporter gene expression than those previously described in which the reporter gene substitutes the viral ORF7a protein (45–47). To generate a rSARS-CoV-2 expressing two reporter genes, mCherry and Nluc, we implemented a similar method and inserted a fusion sequence of mCherry-Nluc, and the PTV-1 2A autoproteolytic peptide, upstream of the SARS-CoV-2 N gene in the BAC containing a full-length copy of the SARS-CoV-2 genome (Fig. 1A), and

rescued rSARS-CoV-2/mCherry-Nluc using our previously described protocol (21, 46). To assess whether mCherry expression could be directly visualized by fluorescence microscopy, Vero E6 cells were mock infected or infected (multiplicity of infection [MOI], 0.01) with rSARS-CoV-2/WT, rSARS-CoV-2/mCherry, rSARS-CoV-2/Nluc, or rSARS-CoV-2/mCherry-Nluc (Fig. 1B). At 24 hours postinfection (hpi), cells were fixed and mCherry expression was directly assessed under a fluorescence microscope, which showed high mCherry fluorescence expression in cells infected with rSARS-CoV-2/mCherry or rSARS-CoV-2/mCherry-Nluc, but not in cells infected with rSARS-CoV-2/WT or rSARS-CoV-2/Nluc (Fig. 1B). Further, viral infection was detected by indirect immunofluorescence microscopy using an anti-N protein 1C7C7 MAb (Fig. 1B). As expected, all Vero E6 cells infected with the different rSARS-CoV-2 mutants, but not mock-infected cells, were positive for the presence of the virus. Expression of mCherry and Nluc reporter genes was also confirmed by Western blotting (Fig. 1C). As expected, mCherry was readily detected in whole-cell lysates from Vero E6 cells infected with rSARS-CoV-2/mCherry or rSARS-Cov-2/mCherry-Nluc but not in those infected with rSARS-CoV-2/WT or rSARS-CoV-2/Nluc or mock infected (Fig. 1C). Likewise, Nluc was detected only in cell extracts from rSARS-CoV-2/Nluc- and rSARS-CoV-2/mCherry-Nluc-infected cells and not in those infected with rSARS-CoV-2/WT or rSARs-CoV-2/mCherry or mock infected (Fig. 1C). A specific band for the viral N protein appeared in all the virus-infected cell extracts, but not in mock-infected Vero E6 cell extracts, all of which showed comparable protein levels of N protein expression (Fig. 1C). The identity of the double reporter-expressing rSARS-CoV-2/mCherry-Nluc was further validated by next-generation sequencing (Fig. 1D). The rSARS-CoV-2/mCherry and rSARS-CoV-2/Nluc were also sequenced as reference controls. We found two nonreference alleles with a frequency greater than 10% in rSARS-CoV-2/mCherry in the viral N (C752T) and envelope E (V5A) proteins (Fig. 1D, top). Likewise, we identified two amino acid changes in the rSARS-CoV-2/Nluc S (H655Y) and E (S6L) proteins (Fig. 1D middle). No amino acid changes were found in rSARS-CoV-2/mCherry-Nluc compared to the reference viral genome (Fig. 1D, bottom), indicating that rSARS-CoV-2/mCherry-Nluc resembles the sequence of rSARS-CoV-2/WT apart from the insertion of the mCherry-Nluc reporter gene fusion and the PTV-1 2A autoproteolytic site.

***In vitro* characterization of rSARS-CoV-2/mCherry-Nluc.** Since the cloning of two reporter genes as a fusion protein could affect viral fitness and/or reporter gene expression, we examined the viral fitness of rSARS-CoV-2/mCherry-Nluc in cultured cells by assessing growth kinetics and compared them to those of rSARS-CoV-2 expressing single reporter gene (e.g., rSARS-CoV-2/mCherry and rSARS-CoV-2/Nluc) or rSARS-CoV-2/WT (Fig. 2A). Vero E6 cells were infected at an MOI of 0.01, and viral titers in cell culture supernatants were quantified at different time points. No significant difference in replication kinetics was found between all the indicated viruses, except for rSARS-CoV-2/Nluc, which replicated slightly slower (Fig. 2A). Conversely, rSARS-CoV-2/mCherry-Nluc reached a high titer of $10^7$ PFU/ml by 24 to 48 hpi like rSARS-CoV-2/WT and rSARS-CoV-2/mCherry, suggesting that the expression of the double reporter fused mCherry-Nluc gene did not affect viral fitness in Vero E6 cells (Fig. 2A). In parallel, Nluc and mCherry expression was evaluated for 96 h by either assessing Nluc activity in cell culture supernatants (Fig. 2B) or by fluorescence microscopy (Fig. 2C). Vero E6 cells were similarly infected (MOI, 0.01), and Nluc activity in cell culture supernatants was quantified at different time points. We found that Nluc activity steadily increased beginning at 12 hpi and peaked at 72 hpi and then decreased at 96 hpi (Fig. 2B). No Nluc activity was detected in cell culture supernatants from mock-infected cells or Vero E6 cells infected with rSARS-CoV-2/WT or rSARS-CoV-2/mCherry (Fig. 2B). Similarly, mCherry expression was detected as early as 12 hpi and increased in a time-dependent matter until 72 hpi (Fig. 2C). At 96 hpi, mCherry expression was lightly reduced, which coincided with the decrease in Nluc activity and viral titers at the same time point most likely due to the cytopathic effect (CPE) caused by viral infection. As expected, no mCherry expression was detected in Vero E6 cells infected with rSARS-CoV-2/WT or rSARS-CoV-2/Nluc or mock infected (not shown). These results suggest that *in vitro*

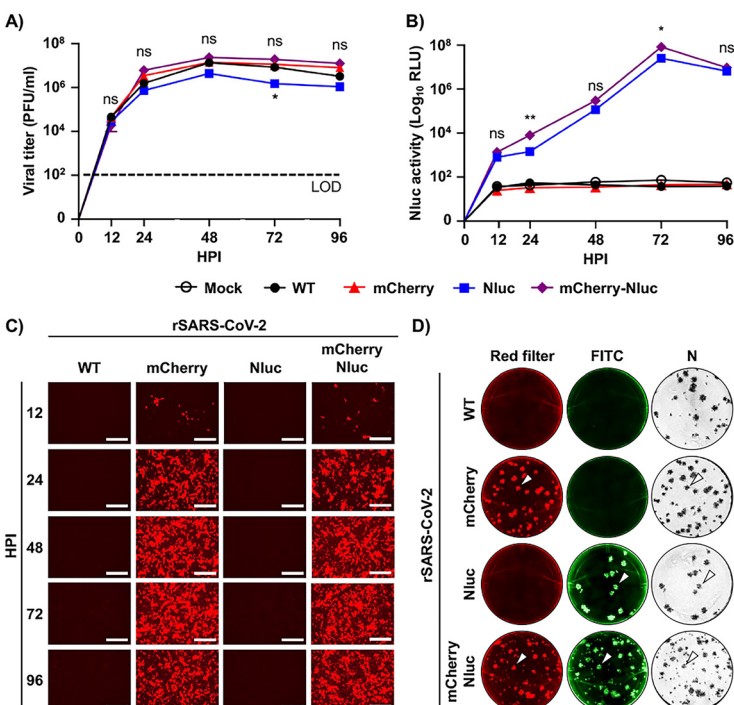

**FIG 2** *In vitro* characterization of the bireporter rSARS-CoV-2/mCherry-Nluc virus. (A) Viral growth kinetics: viral titers (PFU/ml) in the cell culture supernatants of Vero E6 cells ($1.2 \times 10^6$ cells/well, 6-well format, triplicates) infected (MOI, 0.01) with rSARS-CoV-2 WT (WT), rSARS-CoV-2/mCherry (mCherry), rSARS-CoV-2/Nluc (Nluc), or rSARS-CoV-2/mCherry-Nluc (mCherry-Nluc) at the indicated time points postinfection were determined by plaque assay. Data represent the mean values and SD of triplicates. LOD, limit of detection. (B) Nluc activity: Nluc activity in the cell culture supernatants obtained from the experiment in panel A is represented in relative light units (RLU). (C) mCherry expression kinetics: Vero E6 cells ($1.2 \times 10^6$ cells/well, 6-well format, triplicates) were infected (MOI, 0.01) with rSARS-CoV-2 WT, rSARS-CoV-2/mCherry, rSARS-CoV-2/Nluc, or rSARS-CoV-2/mCherry-Nluc and mCherry expression was directly visualized under a fluorescence microscope at the indicated times postinfection. Representative images are shown. Scale bars = 300 $\mu$m. Magnification = ×10. (D) Plaque phenotype: viral plaques from Vero E6 cells ($2 \times 10^5$ cells/well, 24-well plate format, triplicates) infected with rSARS-CoV-2 WT, rSARS-CoV-2/mCherry, rSARS-CoV-2/Nluc, or rSARS-CoV-2/mCherry-Nluc at 3 dpi were observed under a fluorescence imaging system (first column, red filter), fluorescently stained with an antibody against Nluc (second column, FITC), or immunostaining with an antibody against the viral N protein (third column, N). White arrowheads depict the overlapping signal of mCherry fluorescence (left), Nluc bioluminescence (middle), and immunostaining of the viral N protein (right) in Vero E6 cells infected with rSARS-CoV-2/mCherry, rSARS-CoV-2/Nluc, or rSARS-CoV-2/mCherry-Nluc. *, $P < 0.05$; **, $P < 0.01$; ns, not significant.

detection and replication of rSARS-CoV-2/mCherry-Nluc could be monitored and quantified based on the expression of either Nluc (Fig. 2B) or mCherry (Fig. 2C) reporter genes.

Next, plaque assays were conducted to further corroborate that all rSARS-CoV-2/mCherry-Nluc viral particles express both mCherry and Nluc reporter genes and to evaluate plaque phenotype and compared them to those of rSARS-CoV-2 expressing individual reporter genes (rSARS-CoV-2/mCherry and rSARS-CoV-2/Nluc) and rSARS-CoV-2/WT (Fig. 2D). When plaques were examined by fluorescence microscopy, mCherry-positive plaques were detected in cells infected with rSARS-CoV-2/mCherry and rSARS-CoV-2/mCherry-Nluc (Fig. 2D). Then, Nluc-positive plaques were detected using an anti-Nluc specific Ab and FITC-conjugated secondary Ab, which only appeared in cells infected with rSARS-CoV-2/Nluc or rSARS-CoV-2/mCherry-Nluc (Fig. 2D). Importantly, when viral plaques were immunostained with an anti-N protein Ab, they colocalized with mCherry- and/or Nluc-positive plaques (white arrows) in both Vero E6 cells infected with individual reporter-expressing rSARS-CoV-2/mCherry and rSARS-CoV-2/Nluc, as well as in double reporter-expressing rSARS-CoV-2/mCherry-Nluc (Fig. 2D), demonstrating that all rSARS-CoV-2 plaques contained the reporter gene(s). Although the overall plaque size phenotype did not vary between the

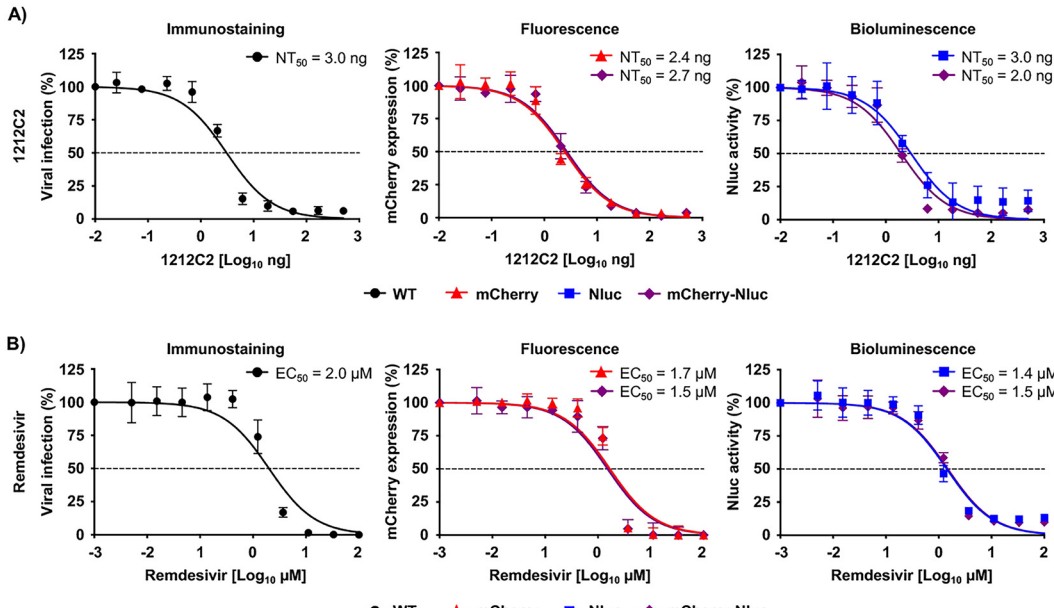

**FIG 3** Bireporter-based microneutralization assay to identify NAbs and antivirals against SARS-CoV-2. (A) A bireporter microneutralization assay to identify NAbs: three-fold serial dilutions of the SARS-CoV-2 1212C2 hMAb (starting concentration of 500 ng) were prepared in postinfection media and incubated with 100–200 PFU/well of rSARS-CoV-2 WT (WT), rSARS-CoV-2/mCherry (mCherry), rSARS-CoV-2/Nluc (Nluc), or rSARS-CoV-2/mCherry-Nluc (mCherry-Nluc) for 1 h at RT. Vero E6 cells (96-well plate format, $4 \times 10^4$ cells/well, quadruplicates) were infected and incubated with the virus-antibody mixture at 37°C for 24 h. Viral neutralization was determined by immunostaining using an anti-N protein MAb (1C7C7) for rSARS-CoV-2/WT (left) or by fluorescence expression for rSARS-CoV-2/mCherry and rSARS-CoV-2/mCherry-Nluc (middle), or bioluminescence for rSARS-CoV-2/Nluc and rSARS-CoV-2/mCherry-Nluc (right) using a microplate reader. The 50% neutralization titer ($NT_{50}$) was calculated using sigmoidal dose-response curves on GraphPad Prism. Viral neutralization was normalized to wells containing infected cells without the 1212C2 hMAb. The dotted line indicates 50% virus inhibition. Data are represented by the mean values and SD of quadruplicates. (B) A bireporter microneutralization assay to assess antivirals: vero E6 cells (96-well plate format, $4 \times 10^4$ cells/well, quadruplicates) were infected with 100 to 200 PFU of rSARS-CoV-2/WT, or reporter viruses expressing mCherry, Nluc, or mCherry-Nluc. After 1 h viral absorption, cells were incubated in postinfection media containing 3-fold serial dilutions of remdesivir (starting concentration of 100 $\mu$M). Viral inhibition was determined by immunostaining using an anti-N protein MAb (1C7C7) for rSARS-CoV-2/WT (left) or by fluorescence expression for rSARS-CoV-2/mCherry and rSARS-CoV-2/mCherry-Nluc (middle), or bioluminescence for rSARS-CoV-2/Nluc and rSARS-CoV-2/mCherry-Nluc (right) using a microplate reader. The 50% effective concentration ($EC_{50}$) was calculated using sigmoidal dose-response curves on GraphPad Prism. Viral inhibition was normalized to wells containing infected cells without remdesivir. The dotted line indicates the 50% virus inhibition. The data are represented by the mean values and SD of quadruplicates.

different viruses, we did observe smaller plaques produced (among the normal-sized plaques) in rSARS-CoV-2/mCherry-Nluc experiments (Fig. 2D).

**A double reporter-based neutralization assay for the identification of SARS-CoV-2 neutralizing antibodies and antivirals.** To demonstrate the feasibility of implementing our rSARS-CoV-2/mCherry-Nluc to identify and characterize neutralizing Abs (NAbs) and antivirals, we developed a double reporter-based microneutralization assay using 1212C2 hMAb (Fig. 3A) and remdesivir (Fig. 3B), which are described and shown to neutralize or inhibit, respectively, SARS-CoV-2 (60, 61). The neutralization activity of 1212C2 was assessed by incubating the hMAb with the indicated rSARS-CoV-2 before infection of Vero E6 cells and quantifying Nluc activity in cell culture supernatants (Fig. 3A, right) and mCherry expression (Fig. 3A, middle) using a microplate reader at 24 hpi. As an internal control, we conducted the microneutralization assay using immunostaining of rSARS-CoV-2/WT, as previously described (Fig. 3A, left) (62). We determined the 50% neutralization concentration ($NT_{50}$) of 1212C2 hMAb using sigmoidal dose-response curves. The $NT_{50}$ values of 1212C2 hMAb against rSARS-CoV-2/mCherry (2.4 ng) and rSARS-CoV-2/mCherry-Nluc (2.7 ng) as determined by fluorescent mCherry expression were similar to those of rSARS-CoV-2/WT (3 ng) using a classical immunostaining assay and those reported with the SARS-CoV-2 WA-1 natural isolate (46, 60). Moreover, $NT_{50}$ values of 1212C2 hMAb against rSARS-CoV-2/Nluc or rSARS-CoV-2/mCherry-Nluc determined by Nluc expression (3.0 and

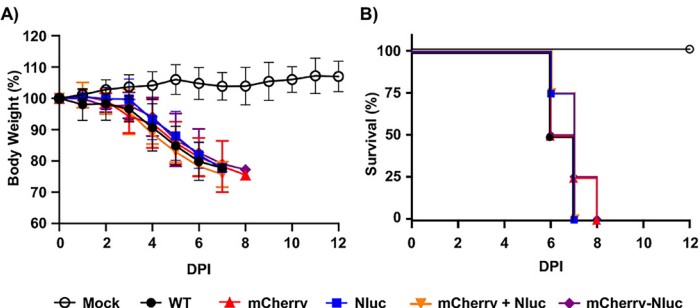

**FIG 4** Virulence of rSARS-CoV-2/mCherry-Nluc in K18 hACE2 transgenic mice: 4- to 6-week-old female K18 hACE2 transgenic mice ($n = 4$) were mock infected or intranasally inoculated with $10^5$ PFU/ mouse of rSARS-CoV-2 WT (WT), rSARS-CoV-2/mCherry (mCherry), rSARS-CoV-2/Nluc (Nluc), or the bireporter rSARS-CoV-2/mCherry-Nluc (mCherry-Nluc). A group of 4- to 6-week-old female K18 hACE2 transgenic mice ($n = 4$) were also coinfected with rSARS-CoV-2/mCherry and rSARS-CoV-2/Nluc (mCherry + Nluc). Body weight loss (A) and survival (B) of mice were monitored for 12 days after viral infection.

2.0 ng, respectively) were also comparable to those of rSARS-CoV-2/WT (3 ng). To determine whether rSARS-CoV-2/mCherry-Nluc could also be used to assess the effectiveness of antivirals, we quantified the Nluc activity (Fig. 3B, right) and mCherry expression (Fig. 3B, middle) in Vero E6 cells infected with the single and double reporter-expressing rSARS-CoV-2 in the presence of serial 3-fold dilutions of remdesivir. As before, we also included rSARS-CoV-2/ WT-infected cells stained with the MAb against the viral N protein as an internal control (Fig. 3B, left). Sigmoidal dose-response curves were developed from reporter expression values and used to calculate the 50% effective concentration ($EC_{50}$). The $EC_{50}$ values of remdesivir against the indicated viruses were similar to each other, regardless of whether the microneutralization assay used immunostaining (rSARS-CoV-2/WT, 2 $\mu$M; left), fluorescence (rSARS-CoV-2/mCherry, 1.7 $\mu$M; rSARS-CoV-2/mCherry-Nluc, 1.5 $\mu$M; middle), or luciferase (rSARS-CoV-2/Nluc, 1.4 $\mu$M; rSARS-CoV-2/mCherry-Nluc, 1.5 $\mu$M; right) (Fig. 3B). Overall, these results demonstrate the feasibility of using the rSARS-CoV-2 expressing both mCherry and Nluc reporter genes to reliably and quickly evaluate the neutralizing and inhibitory properties of NAbs and/or antivirals, respectively, against SARS-CoV-2 based on mCherry and/or Nluc expression, respectively.

**Characterization of rSARS-CoV-2/mCherry-Nluc in K18 hACE2 transgenic mice.** Next, we characterized the pathogenicity and ability of rSARS-CoV-2/mCherry-Nluc to replicate in K18 hACE2 transgenic mice using rSARS-CoV-2 expressing individual fluorescent and bioluminescent reporter genes (rSARS-CoV-2/mCherry and rSARS-CoV-2/ Nluc, respectively) and rSARS-CoV-2/WT as an internal control. One group of mice was infected with a mixture of rSARS-CoV-2/mCherry and rSARS-CoV-2/Nluc. To assess pathogenicity, groups of K18 hACE2 transgenic mice ($n = 4$/group) were mock infected or infected with $10^5$ PFU of the indicated viruses and changes in body weight (Fig. 4A) and survival (Fig. 4B) were monitored for 12 days postinfection (dpi). All mice infected with rSARS-CoV-2 rapidly lost body weight and succumbed to viral infection (Fig. 4A and B, respectively). Most importantly, the virulence of rSARS-CoV-2/mCherry-Nluc was shown to be identical to that of our previously reporter viruses expressing individual mCherry or Nluc (46), or rSARS-CoV-2/WT (62, 63). These data indicate that expression of the fusion of mCherry and Nluc from rSARS-CoV-2/mCherry-Nluc does not result in viral attenuation in the K18 hACE2 transgenic mouse model compared to rSARS-CoV-2/WT.

**Tracking viral dynamics of rSARS-CoV-2/mCherry-Nluc infection and pathogenesis in K18 hACE2 transgenic mice.** Since our rSARS-CoV-2/mCherry-Nluc expresses both fluorescent (mCherry) and luciferase (Nluc) reporter genes, we sought to demonstrate the advantage of using this newly double reporter-expressing rSARS-CoV-2/ mCherry-Nluc to track viral replication in live animals. Thus, K18 hACE2 transgenic mice were mock infected or infected with $10^5$ PFU of the indicated rSARS-CoV-2 reporter viruses intranasally and Nluc was monitored at 1, 2, 4, and 6 dpi (Fig. 5A). In

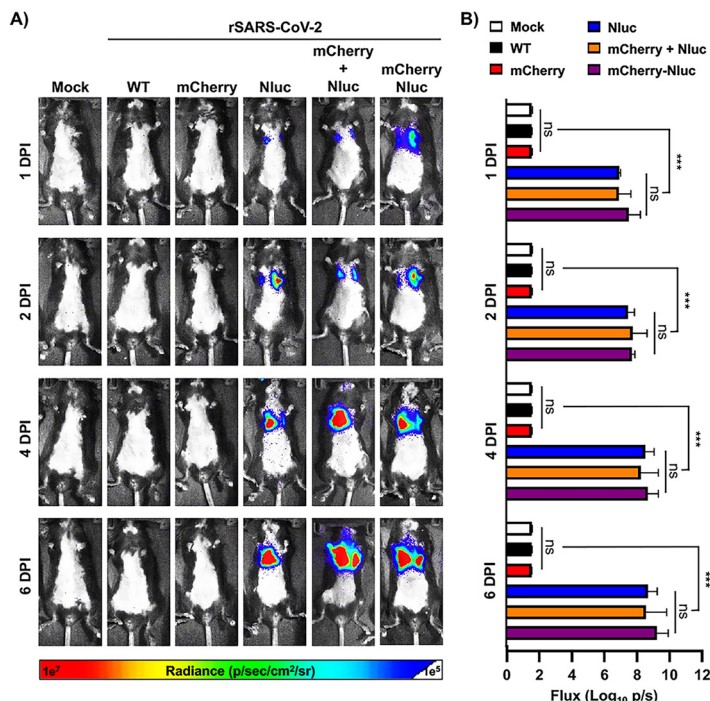

**FIG 5** *In vivo* kinetics of rSARS-CoV-2/mCherry-Nluc in K18 hACE2 transgenic mice: 4- to 6-week-old female K18 hACE2 transgenic mice ($n = 4$) were mock infected or infected intranasally with $10^5$ PFU/mouse of rSARS-CoV-2 WT, rSARS-CoV-2/mCherry, rSARS-CoV-2/Nluc, rSARS-CoV-2/mCherry + rSARS-CoV-2/Nluc, or with rSARS-CoV-2/mCherry-Nluc (mCherry-Nluc). Nluc activity in the whole mouse at the indicated dpi was evaluated with an Ami HT *in vivo* imaging system. Representative images of the same mouse at 1, 2, 4, and 6 dpi are shown (A). Means and SD of the radiance (number of photons per second per square centimeter per steradian [p/second/cm²/sr]) and bioluminescence (total flux [$\log_{10}$ photons per second (p/s)]) over each mouse are shown (B). ***, $P < 0.001$; ns, not significant.

mice infected with rSARS-CoV-2/Nluc or rSARS-CoV-2/mCherry-Nluc, or coinfected at the same time with rSARS-CoV-2/Nluc and rSARS-CoV-2/mCherry, we detected Nluc signal as early as 1 dpi, which increased over time (Fig. 5A). Since IVIS was conducted in the same mouse, viral replication and distribution were followed over time (Fig. 5A) and bioluminescence intensity around the chest area of the mice was measured in flux (Fig. 5B). As expected, Nluc expression increased over time until mice succumbed to SARS-CoV-2 infection, consistent with previous literature, including ours (45). Notably, and as expected based on the IVIS (Fig. 5A), Nluc expression was only readily detected in K18 hACE2 transgenic mice infected with rSARS-CoV-2/Nluc or rSARS-CoV-2/mCherry-Nluc or coinfected with both rSARS-CoV-2/mCherry and rSARS-CoV-2/Nluc (Fig. 5B). No significant differences in flux were observed between the groups of mice infected with the Nluc-expressing rSARS-CoV-2 mutants (Fig. 5B).

As luciferase and fluorescence proteins have different properties and could potentially reveal different readouts as surrogate indicators of viral infection, we next determined and compared Nluc and mCherry expression during infection *in vivo*. Thus, K18 hACE2 transgenic mice ($n = 4$) were mock infected or infected with rSARS-CoV-2/WT, rSARS-CoV-2/mCherry, rSARS-CoV-2/Nluc, or rSARS-CoV-2/mCherry-Nluc or coinfected with rSARS-CoV-2/mCherry and rSARS-CoV-2/Nluc. Then, on 2 and 4 dpi, Nluc activity in the entire mouse (Fig. 6A and B) and mCherry expression of whole lungs (Fig. 6C and D) were determined, including the gross pathology score (Fig. 6E). Like our previous results (Fig. 5), an increase in Nluc expression from 2 to 4 dpi was observed in K18 hACE2 transgenic mice infected with rSARS-CoV-2/Nluc or rSARS-CoV-2/mCherry-Nluc or coinfected with rSARS-CoV-2/mCherry and rSARS-CoV-2/Nluc (Fig. 6A). These results were further confirmed when we determined the flux in the *in vivo* imaged mice (Fig. 6B). After quantifying Nluc expression, the lungs from mock- and rSARS-CoV-2-

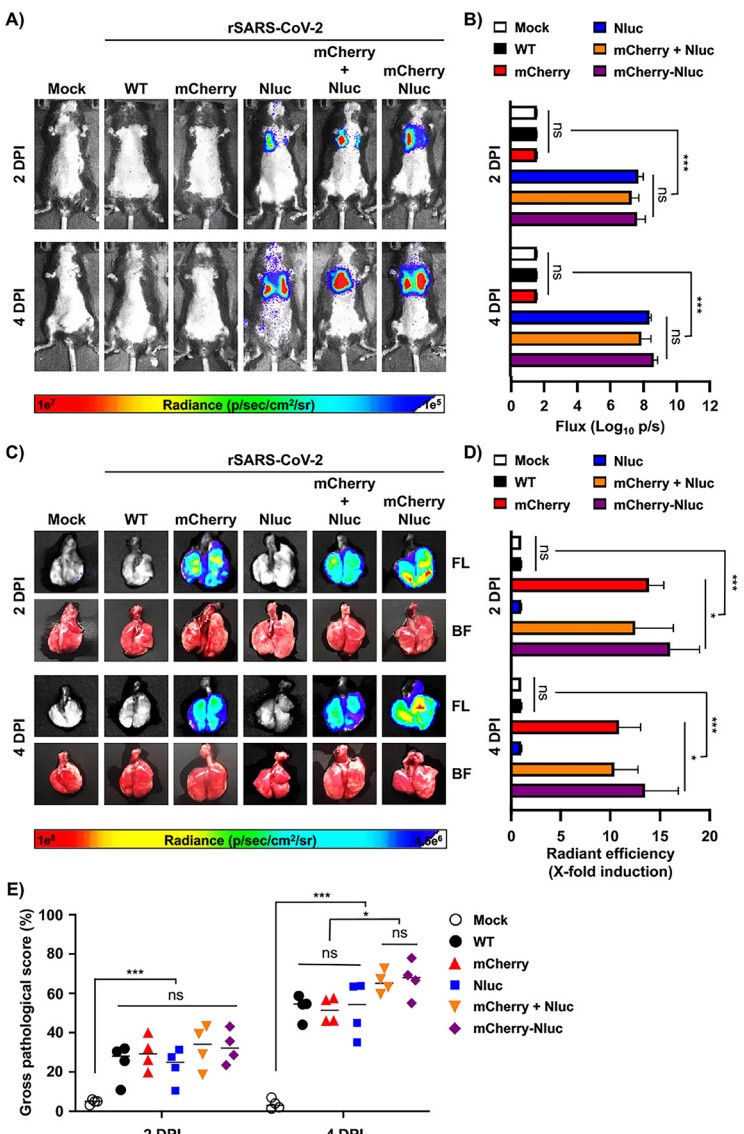

**FIG 6** *In vivo* bioluminescence and *ex vivo* fluorescence in K18 hACE2 transgenic mice infected with rSARS-CoV-2/mCherry-Nluc. (A) *In vivo* Nluc expression: Nluc activity in live mice (*n* = 4) mock infected or infected ($10^5$ PFU/mouse) with rSARS-CoV-2 WT, rSARS-CoV-2/mCherry, rSARS-CoV-2/Nluc, rSARS-CoV-2/mCherry + rSARS-CoV-2/Nluc, or the bireporter rSARS-CoV-2/mCherry-Nluc (mCherry-Nluc) were determined on 2 and 4 dpi using the Ami HT IVIS. A representative image of a mouse per time point is shown. (B) Quantification of Nluc signal: means and SD of the radiance (number of photons per second per square centimeter per steradian [p/second/cm$^2$/sr]) and bioluminescence (total flux [log$_{10}$ photons per second (p/s)]) of mock and infected mice is shown. (C) Ex vivo mCherry expression: excised lungs from mock-infected or infected mice from panel A were monitored for mCherry fluorescent expression (FL, top) and bright field (BF, bottom) at 2 and 4 dpi. Representative lung images from the same mouse used in panel A are shown. (D) Quantification of mCherry expression: the mean values of mCherry signal around the regions of interest were normalized to the autofluorescence of mock-infected lungs at each time point and the fold changes in fluorescence were calculated. (E) Gross pathology score: pathology lesions, consolidation, congestion, and atelectasis, of excised lungs were measured using NIH ImageJ and are represented as percentages of total lung surface area affected. *, $P < 0.05$; ***, $P < 0.001$; ns, not significant.

infected K18 hACE2 transgenic mice were excised and imaged in the IVIS to determine and quantify mCherry expression (Fig. 6C and D, respectively). We only observed detectable levels of mCherry expression in the lungs of K18 hACE2 transgenic mice infected with rSARS-CoV-2/mCherry or rSARS-coV-2/mCherry-Nluc or coinfected with both rSARS-CoV-2/mCherry and rSARS-coV-2/Nluc (Fig. 6C and D). Notably, levels of mCherry expression, like those of Nluc, were comparable in the lungs of K18 hACE2 transgenic mice infected with the double reporter-expressing rSARS-CoV-2/mCherry-

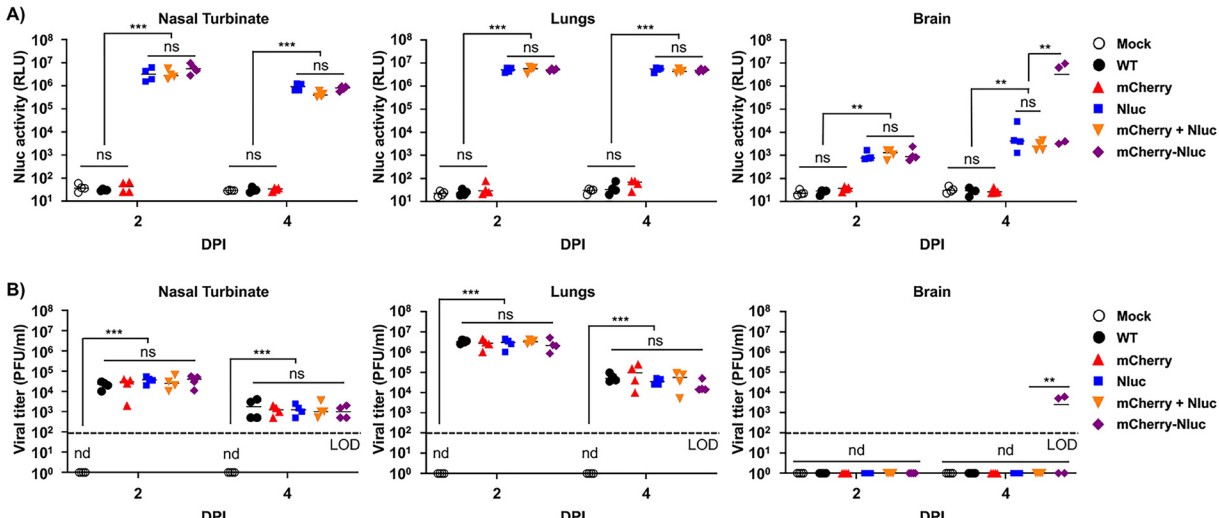

**FIG 7** Nluc activity and viral titers in tissue homogenates from infected K18 hACE2 transgenic mice: the nasal turbinate (left), lungs (middle), and brain (right) of four-to-6-weeks-old female K18 hACE2 transgenic mice ($n = 4$) mock infected or infected intranasally with $10^5$ PFU/mouse of rSARS-CoV-2 WT, rSARS-CoV-2/mCherry, rSARS-CoV-2/Nluc, rSARS-CoV-2/mCherry + rSARS-CoV-2/Nluc, or the bireporter rSARS-CoV-2/mCherry-Nluc were collected after imaging on an Ami HT IVIS on 2 and 4 dpi. After homogenization, Nluc activity (A) and viral titers (B) in tissue homogenates were determined on a microplate reader or by plaque assay, respectively. The results are the mean values and SD. LOD, limit of detection. **, $P < 0.01$; ***, $P < 0.001$; ns, not significant; nd, not detected.

Nluc than those infected with the single rSARS-CoV-2/Nluc or coinfected with rSARS-CoV-2/mCherry and rSARS-CoV-2/Nluc (Fig. 6C and D). Correlating with *in vivo* and *ex vivo* imaging of the lungs, gross lung pathology scores were comparable in all rSARS-CoV-2-infected K18 hACE2 transgenic mice and more significant at 4 dpi (Fig. 6E).

Both Nluc activity and viral titers peaked at 2 dpi in the nasal turbinate of mice infected with rSARS-CoV-2/Nluc or rSARS-CoV-2/mCherry-Nluc or coinfected with rSARS-CoV-2/mCherry and rSARS-CoV-2/Nluc (Fig. 7A and B, left) However, in the lungs, Nluc activity remained the same at 2 and 4 dpi, while viral titers decreased at 4 dpi compared to 2 dpi (Fig. 7A and B, middle). Nluc activity in brain homogenates was only evident in the samples from mice infected with rSARS-CoV-2/Nluc, rSARS-CoV-2/mCherry-Nluc, or both rSARS-CoV-2/mCherry and rSARS-CoV-2/Nluc, and signals increased in a time-dependent matter (Fig. 7A, right). Consistent with previous studies, we were only able to detect SARS-CoV-2 in the brain of two of the four mice infected with rSARS-CoV-2/mCherry-Nluc at 4 dpi (Fig. 7B) (45). Altogether, these findings demonstrate the feasibility of assessing viral infection *in vivo* in the entire mouse by bioluminescence (Nluc) and *ex vivo* in the lungs of infected mice by fluorescence (mCherry) with our double reporter-expressing rSARS-CoV-2/mCherry-Nluc and that the mCherry-Nluc fusion does not have a significant impact in the pathogenesis and replication of the virus in K18 hACE2 transgenic mice, showing similar levels of Nluc or mCherry reporter gene expression than those of rSARS-CoV-2 expressing individual bioluminescence or fluorescence proteins. Notably, viral titers of rSARS-CoV-2 mCherry-Nluc in the nasal turbinate and lungs were comparable to those of a rSARS-CoV-2/WT.

**Assessment of SARS-CoV-2 infection and transmission in golden Syrian hamsters.** To demonstrate the feasibility of using our double reporter rSARS-CoV-2/mCherry-Nluc to assess viral replication and transmission, golden Syrian hamsters ($n = 4$) were mock infected or infected with $10^5$ PFU/hamster of rSARS-CoV-2/mCherry-Nluc. The day after infection, noninfected naive contact hamsters were placed in the same cage as infected hamsters. On 2, 4, and 6 dpi, Nluc expression in the entire hamsters was evaluated by IVIS, like in our previous studies using K18 hACE2 transgenic mice. Infected hamsters presented detectable levels of Nluc expression in both the nasal turbinate and lungs at 2 and 4 dpi that decreased at 6 dpi (Fig. 8A). In contrast, contact hamsters had little to no Nluc signal on 2 dpi, which drastically increased on 4 and 6 dpi (Fig. 8A). The temporal and spatial differences in Nluc signal between

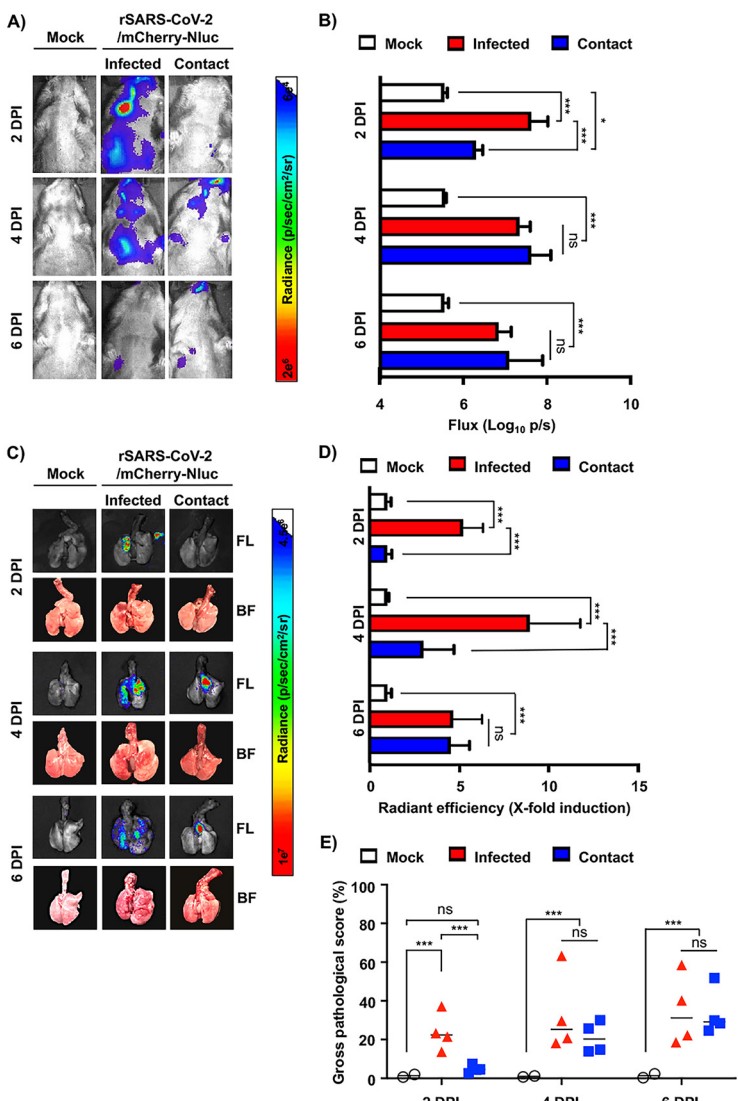

**FIG 8** *In vivo* bioluminescence and *ex vivo* fluorescence in golden Syrian hamsters infected with rSARS-CoV-2/mCherry-Nluc. (A) *In vivo* Nluc expression: Nluc activity in 4- to 6-week-old female golden Syrian hamsters ($n = 4$) mock infected or infected with $10^5$ PFU/hamster of rSARS-CoV-2/mCherry-Nluc were determined on 2, 4, and 6 dpi using the Ami HT IVIS. Contact animals were exposed to infected animals 1 dpi. A representative image of a hamster per time points and experimental condition is shown. (B) Quantification of Nluc signal: means and SD of the radiance (number of photons per second per square centimeter per steradian [p/second/cm²/sr]) and bioluminescence (total flux [$\log_{10}$ photons per second (p/s)]) were quantified from whole hamsters after IVIS imaging. (C) Ex vivo mCherry and Nluc expression: excised lungs from mock-infected or infected golden Syrian hamsters from panel A were monitored for mCherry fluorescence (FL, top), Nluc signal (Nluc, middle), and bright field (BF, bottom) at 2, 4, and 6 dpi. Representative lung images from the same hamster used in panel A are shown. (D) Quantification of mCherry expression: the mean values of mCherry signal around the regions of interest were normalized to the autofluorescence of mock-infected lungs at each time point and the fold changes in fluorescence were calculated. (E) Gross pathology score: pathological lesions, consolidation, congestion, and atelectasis, of excised lungs were measured using NIH ImageJ and are represented as percentages of total lung surface area affected. *, $P < 0.05$; ***, $P < 0.001$; ns, not significant.

originally infected and contact hamsters are most likely due to the route of transmission/infection and the time frame in which the contact hamsters were exposed to the originally infected hamsters. These initial IVIS results were further confirmed by quantification of bioluminescence in hamsters (Fig. 8B), which showed a decrease in flux in infected hamsters from 2 to 6 dpi and an increase from 2 to 4 dpi and then decreased on 6 dpi in contact hamsters (Fig. 8B). Subsequently, lungs were excised and imaged in the IVIS for Nluc and mCherry expression (Fig. 8C). Nluc and mCherry levels of

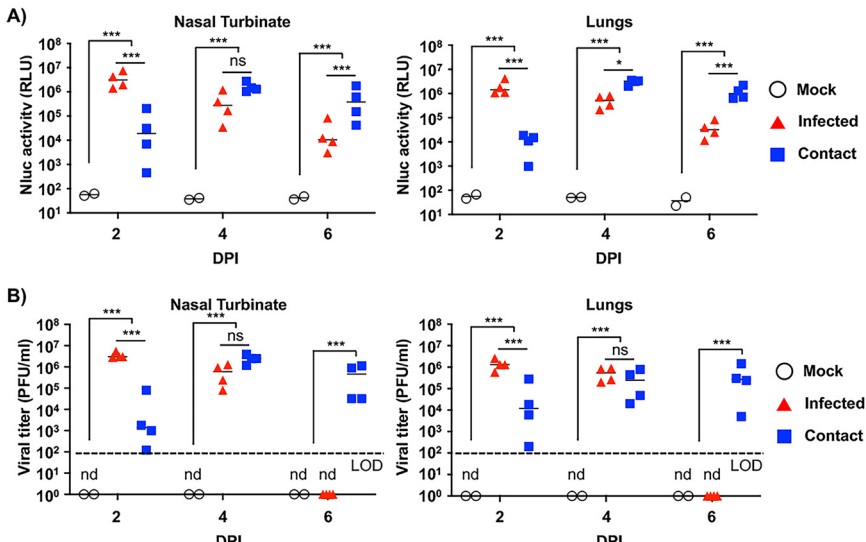

**FIG 9** Nluc activity and viral titers in golden Syrian hamster tissue homogenates infected with rSARS-CoV-2/mCherry-Nluc: the nasal turbinate (left) and lungs (right) of 4- to 6-week-old female golden Syrian hamsters ($n = 4$) mock infected or infected intranasally with $10^5$ PFU/hamster of rSARS-CoV-2/mCherry-Nluc were collected after imaging on an Ami HT IVIS at 2 and 4 dpi. In addition, after 24 hpi, contact golden Syrian hamsters ($n = 4$, contact) were added to the cages of infected animals. After homogenization, Nluc activity (A) and viral titers (B) in tissue homogenates were determined on a microplate reader or by plaque assay, respectively. Results are the means and SD. LOD, limit of detection. *, $P < 0.05$; ***, $P < 0.001$; ns, not significant; nd, not detected.

expression correlate between them and with that observed by IVIS in the whole hamster, revealing a time-dependent effect in reporter expression in both infected and contact hamsters that correlate with the levels observed by IVIS (Fig. 8D). Gross pathology scores in the lungs of infected and contact hamsters were determined from brightfield images with increased scores over time in both infected and contact animals (Fig. 8E).

Finally, nasal turbinate and lungs from mock and originally infected or contact hamsters were processed to determine Nluc activity (Fig. 9A) and viral titers (Fig. 9B). In the nasal turbinate of originally infected hamsters, both Nluc activity and viral titers decreased from 2 to 6 dpi, while in contact hamsters, a time-dependent increase was observed (Fig. 9A and B, respectively). A similar trend between Nluc signal and viral titers was observed in the lungs of originally infected and contact hamsters. Notably, Nluc signal and viral titers from both groups of infected and contact hamsters correlated with bioluminescence intensity from whole animals and excised lungs (Fig. 8). Based on these results with rSARS-CoV-2/mCherry-Nluc, viral infection can be monitored in hamsters solely using reporter expression that correlates well with levels of viral replication. Moreover, rSARS-CoV-2 transmission from originally infected to contact hamsters can be easily tracked *in vivo* in the whole animal or *ex vivo* in the lungs (Fig. 8), and results correlate with those of viral replication (Fig. 9).

## DISCUSSION

Replication-competent, reporter-expressing, recombinant viruses have been previously shown to represent an excellent approach to study, among others, viral infection, replication, pathogenesis, and transmission (21–31). We and others have described the feasibility of generating rSARS-CoV-2 expressing reporter genes encoding either fluorescent or luciferase proteins (24, 38, 45, 46, 48). These reporter-expressing replication-competent rSARS-CoV-2 can be used to assess the prophylactic activity of vaccines and/or the therapeutic potential of NAbs or antivirals (47, 62). Moreover, these rSARS-CoV-2 expressing fluorescent or luciferase proteins represent an excellent option to study the biology of SARS-CoV-2 in cultured cells and/or in validated small animals of

infection (24, 46, 47). Moreover, we have described a new approach to express reporter genes from the locus of the viral N protein using a 2A autoproteolytic system where levels of fluorescent or luciferase expression are higher than those where the reporter gene substitutes the viral ORF7a (45, 46). This new 2A strategy to express the reporter gene from the viral N protein locus does not require deleting the viral ORF7a (45, 46). However, these previously described reporter-expressing replication-competent rSARS-CoV-2 only express a single fluorescent or a luciferase protein; therefore, their experimental applications are limited to the properties of one specific reporter gene and available equipment (45, 46).

Using our previously described BAC-based reverse genetics (21, 58), we generated a rSARS-CoV-2 expressing a fusion of the fluorescent mCherry protein to the bioluminescence Nluc protein (rSARS-CoV-2/mCherry-Nluc) upstream of the viral N protein separated by the PTV-1 2A autoproteolytic cleavage site, thereby allowing separate expression of the mCherry-Nluc fusion and the viral N protein (45, 46). The expression of both reporter genes was validated by fluorescence microscopy (mCherry) or by luciferase activity with a microplate reader (Nluc). We further confirmed reporter expression by Western blotting, where a specific band was detected for the mCherry-Nluc fusion polyprotein. In cell culture, rSARS-CoV-2/mCherry-Nluc displayed growth kinetics similar to those of rSARS-CoV-2 expressing individual mCherry (rSARS-CoV-2/mCherry) or Nluc (rSARS-CoV-2/Nluc) or a rSARS-CoV-2 lacking reporter genes (rSARS-CoV-2/WT). Likewise, the plaque phenotype of the novel rSARS-CoV-2/mCherry-Nluc was similar in size to that of rSARS-CoV-2/mCherry, rSARS-CoV-2/Nluc, or rSARS-CoV-2/WT but only rSARS-CoV-2/mCherry-Nluc had detectable levels of expression of both reporter genes. Importantly, levels of mCherry or Nluc reporter expression correlated with the levels of viral replication, further supporting the concept of reporter genes being a valid surrogate to study viral infection.

Based on the advantages of using a rSARS-CoV-2 expressing both a fluorescent and a bioluminescence protein over those expressing either fluorescence or luciferase reporter genes, we developed a bireporter-based microneutralization assay to identify and characterize NAbs as well as antivirals. Importantly, $NT_{50}$ (NAbs) and $EC_{50}$ (antivirals) values obtained in bireporter-based microneutralization assays using either fluorescence or luciferase signal were comparable to those obtained with rSARS-CoV-2 expressing individual mCherry (rSARS-CoV-2/mCherry) or Nluc (rSARS-CoV-2/Nluc) reporter genes or rSARS-CoV-2/WT and those reported previously in the literature (24, 45, 46, 48). Overall, the bireporter rSARS-CoV-2 represents an excellent option in circumstances where fluorescence or luciferase is negated by the properties of an antiviral drug (such as fluorescing chemical entities in certain small molecule compounds) or the host cell being studied (46, 62). Moreover, although in this article the bireporter-based microneutralization assay was performed in 96-well plates, it can be easily adapted to a 384-well format for high-throughput screenings (HTS) to identify NAbs or antivirals using a double reporter screening approach based on the expression of both fluorescent mCherry and luciferase Nluc. In this instance, having two reporters allows HTS to have the option to use either reporters or both, to further validate neutralization results.

One of the major limitations of recombinant viruses expressing reporter genes is their potential attenuation *in vivo* (39). To assess whether the expression of mCherry fused to Nluc affected SARS-CoV-2 replication *in vivo*, we infected K18 hACE2 transgenic mice and golden Syrian hamsters with rSARS-CoV-2/mCherry-Nluc. Despite encoding a fusion of two reporter genes from the locus of the viral N protein, rSARS-CoV-2/mCherry-Nluc displayed similar virulence as determined by changes in body weight and survival in K18 hACE2 transgenic mice as rSARS-CoV-2 expressing individual fluorescent (rSARS-CoV-2/mCherry) or luciferase genes (rSARS-CoV-2/Nluc) or rSARS-CoV-2/WT. Importantly, we traced viral infection in the same animal for 6 days based on Nluc expression. We were able to detect both luciferase and fluorescent expression in the lungs of mice infected with rSARS-CoV-2/mCherry-Nluc, and mCherry or Nluc expression levels in the lungs of infected mice were comparable to those observed in mice infected with rSARS-CoV-2/mCherry or rSARS-CoV-2/Nluc or coinfected

with both rSARS-CoV-2/mCherry and rSARS-CoV-2/Nluc or rSARS-CoV-2/WT. Notably, rSARS-CoV-2/mCherry-Nluc replicated in the nasal turbinate, lungs, and brain of infected K18 hACE2 transgenic mice to levels comparable to recombinant viruses expressing individual reporter genes (rSARS-CoV-2/mCherry or rSARS-CoV-2/Nluc) and those of rSARS-CoV-2/WT. Similar results were also observed in the golden Syrian hamster model of SARS-CoV-2 infection and transmission (21, 60). Importantly, in the case of hamsters, we were able to track viral infection and transmission in infected and contact hamsters, respectively, demonstrating the feasibility of using our double reporter-expressing rSARS-CoV-2/mCherry-Nluc in transmission studies in hamsters.

In summary, we have generated a rSARS-CoV-2 expressing simultaneously two reporter genes that is suitable for multiple experimental applications currently not available with the use of rSARS-CoV-2 expressing a single fluorescent or luciferase reporter gene. This rSARS-CoV-2/mCherry-Nluc virus is, to our knowledge, the first replication-competent rSARS-CoV-2 stably expressing two reporter genes. With rSARS-CoV-2/mCherry-Nluc, mCherry could be used to identify infected cells *in vitro* while Nluc represents a better option to provide quantified levels of infection. In animal studies, mCherry is a superior option for *ex vivo* imaging and the identification of infected cells (45). Moreover, by combining two recombinant viruses expressing two different fluorescent proteins, one could evaluate the antiviral or neutralizing activity of antivirals or antibodies, respectively, against multiple viruses by looking at different fluorescent expression (47). In contrast, Nluc is the only viable option for *in vivo* imaging using entire animals (45). Importantly, both reporter genes can be used as valid surrogates of viral infection since their levels of expression correlate with those of viral replication, demonstrating the feasibility of using this novel bireporter-expressing rSARS-CoV-2 to study the biology of SARS-CoV-2 *in vitro* and/or *in vivo*. The feasibility of generating rSARS-CoV-2 expressing a fusion of two reporter genes demonstrates the plasticity of the viral genome to express large ORFs from the locus of the viral N protein. Moreover, the robust levels of reporter gene expression obtained using this 2A autoproteolytic approach and the feasibility of expressing foreign genes without the need of removing a viral protein (e.g., ORF7a) represent an ideal option for the use of rSARS-CoV-2/mCherry-Nluc to study viral infection, pathogenesis, and transmission, including newly identified VoC.

## MATERIALS AND METHODS

**Biosafety and ethics statement.** *In vitro* and *in vivo* experiments involving infectious rSARS-CoV-2 were conducted in a biosafety level 3 (BSL3) laboratory at Texas Biomedical Research Institute. Experimental procedures involving cell culture and animal studies were approved by the Texas Biomedical Research Institute Biosafety and Recombinant DNA Committees (BSC and RDC, respectively) and the Institutional Animal Care and Use Committee (IACUC).

**Cells and viruses.** African green monkey kidney epithelial cells (Vero E6; CRL-1586) were propagated and maintained in Dulbecco's modified Eagle's medium (DMEM; Corning) supplemented with 5% fetal bovine serum (FBS; VWR) and 1% PSG (100 U/ml penicillin, 100 $\mu$g/ml streptomycin, and 2 mM L-glutamine; Corning) at 37°C with 5% $CO_2$.

Recombinant (r)SARS-CoV-2 was generated based on the whole genomic sequence of the USA-WA1/2020 (WA-1) strain (accession no. MN985325) (21, 45) using a previously described bacterial artificial chromosome (BAC)-based reverse genetics system (21–23, 58, 59). Viral titers (PFU/ml) were determined by plaque assay in Vero E6 cells.

**Rescue of recombinant double reporter-expressing SARS-CoV-2.** A BAC plasmid was used for the rescue of rSARS-CoV-2 expressing mCherry and Nanoluciferase (Nluc), referred to as rSARS-CoV-2/mCherry-Nluc, as previously described (45). Briefly, a fused version of mCherry and Nluc was inserted in front of the viral N protein gene along with a porcine teschovirus 1 (PTV-1) 2A autocleavage site, within the pBeloBAC11 plasmid (NEB) containing the whole genomic sequence of SARS-CoV-2 WA-1 strain. We chose mCherry because red fluorescent proteins are more readily detectable in biological tissues, enabling lower absorbance and scattering of light, as well as less autofluorescence (47, 64–66). We selected Nluc due to its small size, ATP independence, and greater sensitivity and brightness compared with other luciferases (45, 48, 49, 67). Vero E6 cells (1.2 $\times$ 10$^6$ cells/well, 6-well plate format, triplicates) were transfected in suspension with 4.0 $\mu$g/well of SARS-CoV-2/mCherry-Nluc BAC plasmid using Lipofectamine 2000 (Thermo Fisher Scientific). The transfection medium was changed to postinfection media (DMEM containing 2% FBS and 1% PSG) after 24 h, and cells were split and seeded into T75 flasks 2-days posttransfection. After 3 days, viral rescues were detected by fluorescence microscopy, and cell culture supernatants were collected, labeled as P0, and stored at −80°C. After viral titration, P1 viral

stocks were generated by infecting fresh Vero E6 cells at a low multiplicity of infection (MOI) of 0.0001 for 3 days and following stored at −80°C.

**Reverse transcription-PCR.** Total RNA was extracted from rSARS-CoV-2/mCherry-Nluc-infected (MOI, 0.01) Vero E6 cells ($1.2 \times 10^6$ cells/well, 6-well format) using TRIzol reagent (Thermo Fisher Scientific) based on the manufacturer's instructions. The viral genome between 27,895 and 29,534 nucleotides based on the SARS-CoV-2 WA-1 strain was reverse transcription (RT)-PCR amplified using Super Script II Reverse transcriptase (Thermo Fisher Scientific) and Expanded High Fidelity PCR system (Sigma-Aldrich). Amplified DNA products were separated on a 0.7% agarose gel, purified using a Wizard Genomic DNA Purification kit (Promega), and sent for Sanger sequencing (ACGT). Primer sequences are available upon request.

**Deep sequencing.** The RNA sequencing library was prepared with a KAPA RNA HyperPrep kit, involving 100 ng of viral RNA and 7 mM adaptor, and was subjected to a 45-min adaptor ligation incubation and 6 cycles of PCR. An Illumina Hiseq X was used to sequence all samples and raw sequencing reads were trimmed and filtered using Trimmomatic V0.32 (68, 69). Bowtie2 V2.4.1 (70) and MosDepth V0.2.6 (71) were used to map sequence reads and quantify genome coverage to reference SARS-CoV-2-WA1/2020 viral genome (MN985325.1), respectively. LoFreq V2.1.3.1 (72) was used to determine low-frequency variants and eliminate sites that were less than 100 read depth or less than 1% allele frequencies.

**Immunofluorescence assays.** Vero E6 cells ($1.2 \times 10^6$ cells/well, 6-well format, triplicates) were mock infected or infected (MOI, 0.01) with rSARS-CoV-2/WT, rSARS-CoV-2/mCherry, rSARS-CoV-2/Nluc, or rSARS-CoV-2/mCherry-Nluc. At 24 h postinfection (hpi), cells were fixed in 10% neutral buffered formalin at 4°C overnight and permeabilized using 0.5% Triton X-100 in phosphate-buffered saline (PBS) for 10 min at room temperature (RT). Cells were washed with PBS, blocked with 2.5% bovine albumin serum (BSA) in PBS for 1 h, and then incubated with 1 $\mu$g/ml of SARS-CoV anti-N monoclonal antibody (MAb) 1C7C7 in 1% BSA at 4°C overnight. Cells were washed with PBS and incubated with a fluorescein isothiocyanate (FITC)-conjugated goat anti-mouse IgG (Dako; 1:200). Cell nuclei were stained with 4′,6′-diamidino-2-phenylindole (DAPI; Research Organics). Representative images (20×) were acquired using an EVOS M5000 imaging system (Thermo Fisher Scientific).

**SDS-PAGE and Western blot.** Cell lysates were prepared from either mock- or virus-infected (MOI, 0.01) Vero E6 cells ($1.2 \times 10^6$ cells/well, 6-well format) after 24 hpi using passive lysis buffer (Promega) based on the manufacturer's instructions. After centrifugation ($12,000 \times g$) at 4°C for 30 min, proteins were separated with 12% SDS-PAGE and transferred to nitrocellulose membranes. Membranes were blocked for 1 h with 5% dried skim milk in 0.1% Tween 20 PBS (T-PBS) and incubated at 4°C overnight with the following specific primary MAbs or polyclonal antibodies (PAbs): N (mouse MAb 1C7C7), mCherry (rabbit Pab; Raybiotech), and Nluc (rabbit Pab, Promega). Then, membranes were incubated at 37°C for 1 h with goat anti-mouse IgG StarBright Blue 520 or anti-rabbit IgG Starbright Blue 700 (Bio-Rad) secondary antibodies. Tubulin was used as a loading control using an anti-tubulin hFAB rhodamine antibody (Bio-Rad). Proteins were detected using a ChemiDoc MP imaging system (Bio-Rad).

**Plaque assay.** Vero E6 cells ($2 \times 10^5$ cells/well, 24-well plate format, triplicates) were infected with 25 to 50 PFU of rSARS-CoV-2/WT, rSARS-CoV-2/mCherry, rSARS-CoV-2/Nluc, or rSARS-CoV-2/mCherry-Nluc for 1 h, overlaid with postinfection media containing 0.6% agar (Oxoid), and incubated at 37°C in a 5% $CO_2$ incubator. At 72 hpi, cells were fixed in 10% neutral buffered formalin at 4°C overnight, and then mCherry-positive plaques were visualized using a ChemiDoc MP imaging system (Bio-Rad). Afterward, cells were permeabilized in T-PBS for 10 min at RT, blocked in 2.5% BSA in PBS for 1 h, and incubated with specific primary MAb or PAb against the viral N protein (mouse MAb 1C7C7) or Nluc (rabbit PAb). To detect Nluc-positive viral plaques, cells were stained with a FITC-conjugated goat anti-rabbit IgG (Dako; 1:200) and visualized using a ChemiDoc MP imaging system (Bio-Rad). Next, viral plaques were stained with an anti-mouse Vectastain ABC kit and DAB HRP Substrate kit (Vector laboratories) following the manufacturers' recommendations.

**Viral growth kinetics.** Vero E6 cells ($1.2 \times 10^6$ cells/well, 6-well plate format, triplicates) were infected (MOI, 0.01) with rSARS-CoV-2/WT, rSARS-CoV-2/mCherry, rSARS-CoV-2/Nluc, or rSARS-CoV-2/mCherry-Nluc. After 1 h of adsorption, cells were washed with PBS and incubated at 37°C in postinfection media. Viral titers in cell culture supernatants at each of the indicated time points (12, 24, 48, 72, and 96 hpi) were determined by plaque assay as described above. At each time point, mCherry expression was visualized with an EVOS M5000 imaging system. Nluc activity in the cell culture supernatants at the same times postinfection was quantified using a microplate reader and a Nano-Glo Luciferase Assay system (Promega) following the manufacturer's recommendations. Mean values and standard deviation (SD) were calculated with Microsoft Excel software.

**Reporter-based microneutralization and antiviral assays.** Microneutralization and antiviral assays were performed as previously described (47, 62). Human MAb 1212C2 (60) against the Spike protein receptor-binding domain (RBD) of SARS-CoV-2 was serially diluted (3-fold) in postinfection media (starting concentration of 500 ng), combined with 100 to 200 PFU/well of rSARS-CoV-2/WT, rSARS-CoV-2/mCherry, rSARS-CoV-2/Nluc, or rSARS-CoV-2/mCherry-Nluc and incubated at RT for 1 h. Then, Vero E6 cells ($4 \times 10^4$ cells/well, 96-well plate format, quadruplicates) were infected with the antibody-virus mixture and incubated at 37°C in a 5% $CO_2$ incubator. Cells infected with rSARS-CoV-2/WT were overlaid with 1% Avicel as previously described (62). Nluc activity in cell culture supernatants of cells infected with rSARS-CoV-2/Nluc or rSARS-CoV-2/mCherry-Nluc was quantified at 24 hpi using Nano-Glo luciferase substrate as per the manufacturer's instructions, and a Synergy LX microplate reader and analyzed using a Gen5 data analysis software (Bio-Tek). To measure the mCherry signal, cells infected with rSARS-CoV-2/mCherry or rSARS-CoV-2/mCherry-Nluc were fixed in 10% neutral buffered formalin overnight and washed with PBS before being quantified in a Synergy LX microplate reader. For cells infected with rSARS-CoV-2/WT, plaques were detected using the anti-N MAb 1C7C7 as indicated above and quantified using an ImmunoSpot Analyzer (CTL). Total viral infection (100%) was determined from the number of plaques, fluorescence, and luciferase values obtained from virus-infected cells without the 1212C2 hMAb. Viral infection means and SD values were calculated from quadruplicate individual wells of three independent

experiments with Microsoft Excel software. Nonlinear regression curves and 50% neutralization titer ($NT_{50}$) values were determined using GraphPad Prism Software (San Diego, CA, USA, V. 8.2.1).

Inhibition of SARS-CoV-2 in antiviral assays was conducted as previously described (47, 62). Briefly, Vero E6 cells ($4 \times 10^4$ cells/well, 96-well plate format, quadruplicates) were infected with 100 to 200 PFU/well of rSARS-CoV-2/WT, rSARS-CoV-2/mCherry, rSARS-CoV-2/Nluc, or rSARS-CoV-2/mCherry-Nluc and incubated at 37°C for 1 h. Afterward, the virus inoculum was removed and replaced with postinfection media containing 3-fold serial dilutions of remdesivir (starting concentration of 100 $\mu$M), and cells were incubated at 37°C in a 5% $CO_2$ incubator. Cells infected with rSARS-CoV-2/WT were overlaid with 1% Avicel as previously described (62). After 24 hpi, Nluc activity from cell culture supernatants infected with rSARS-CoV-2/Nluc or rSARS-CoV-2/mCherry-Nluc was determined using Nano-Glo luciferase substrate and a Synergy LX microplate reader. For cells infected with rSARS-CoV-2/mCherry or rSARS-CoV-2/mCherry-Nluc, mCherry expression was quantified in a Synergy LX microplate reader. Lastly, rSARS-CoV-2/WT was detected using the anti-N MAb 1C7C7 and quantified using an ImmunoSpot Analyzer (CTL). Total viral infection (100%) was calculated from the number of plaques, fluorescence, and luciferase values of infected cells in the absence of remdesivir. Means and SD values were calculated from quadruplicates from three independent experiments with Microsoft Excel software. The 50% effective concentration ($EC_{50}$) was calculated by sigmoidal dose-response curves on GraphPad Prism (San Diego, CA, USA, version 8.2.1).

**Mice experiments.** Female 4- to 6-week-old K18 hACE2 transgenic mice [B6.Cg-Tg(K18-ACE2)2Prlmn/J; The Jackson Laboratory] were maintained in the animal care facility at Texas Biomedical Research Institute under specific pathogen-free conditions. For viral infections, groups of mice were anesthetized with gaseous isoflurane and inoculated intranasally with the indicated viruses. A separate group of K18 hACE2 transgenic mice were also mock infected with PBS and served as a negative control.

For body weight and survival studies, K18 hACE2 transgenic mice ($n = 4$) were intranasally infected with $10^5$ PFU/mouse of the indicated viruses and monitored daily for body weight loss and survival to access morbidity and mortality, respectively, for 12 days. Mice that were below 75% of their initial body weight were considered to have reached their experimental end point and were humanly euthanized.

*In vivo* bioluminescence imaging of live mice ($n = 4$) was conducted with an Ami HT *in vivo* imaging system (IVIS; Spectral Instruments) at 1, 2, 4, and 6 days postinfection (dpi). At each time point, mice were anesthetized with isoflurane and retro-orbitally injected with 100 $\mu$L of Nano-Glo luciferase substrate diluted by 1:10 in PBS. Mice were immediately placed in an isolation chamber and imaged using the Ami HT IVIS. Radiance within the region of interest (ROI) of each mouse was analyzed using the Aura software (Spectral Instruments), and total flux values (protons/s) were normalized to the background signal of mock-infected control.

To access fluorescence expression in the lungs and to determine viral titers, a separate cohort of mice ($n = 4$) were similarly infected with the indicated recombinant viruses and were humanely euthanized at 2 and 4 dpi after *in vivo* imaging. Lungs were surgically excised and washed in PBS, and fluorescent and brightfield images were obtained using an Ami HT IVIS and an iPhone 6s (Apple), respectively. The fluorescence signal (radiance efficiency) around the ROI of the lungs was quantified using the Aura software and mean values were normalized to the autofluorescence of mock-infected lungs. The macroscopic pathology score was determined in a blinded manner by a certified pathologist from brightfield images of the lungs, in which the percentage of the total surface area of lungs affected by consolidation, lesions, congestion, and/or atelectasis was quantified with NIH ImageJ software as previously described (21, 73). Nasal turbinate and brains were also collected, and tissues were individually homogenized in 1 mL of PBS using a Precellys tissue homogenizer (Bertin Instruments). Tissue homogenates were centrifuged at 12,000 $\times$ $g$ at 4°C for 5 min to pellet cell debris, and supernatants were collected. Viral titers were determined by plaque assay and immunostaining as described above. Nluc activity in the tissue homogenates was determined using Nano-Glo luciferase substrate kit and a Synergy LX microplate reader.

**Hamster experiments.** Female 4- to 6-week-old golden Syrian hamsters (*Mesocricetus auratus*) were purchased from Charles River Laboratories and maintained in the animal care facility at Texas Biomedical Research Institute under specific pathogen-free conditions. For viral infections, hamsters were anesthetized with isoflurane and intranasally infected with rSARS-CoV-2/mCherry-Nluc. One day later, infected hamsters were transferred to cages containing contact naive hamsters. A separate group of hamsters were also mock infected with PBS and served as a negative control.

*In vivo* bioluminescence imaging of live hamsters ($n = 4$) was conducted with an Ami HT IVIS on 2, 4, and 6 dpi. Hamsters were anesthetized with gaseous isoflurane in an isolation chamber, and Nano-Glo luciferase substrate was diluted 1:10 in PBS and retro-orbitally injected into each animal (200 $\mu$L). Immediately after, hamsters were secured in the isolation chamber and imaged with an Ami HT IVIS and bioluminescence analyses were performed. The total flux values were obtained around the ROI of each hamster and normalized to mock-infected hamsters. Next, hamsters were euthanized, and mCherry expression in excised lungs was imaged in an Ami HT IVIS. The Aura software was used to determine the radiant efficiency of the ROI. Fluorescence signals obtained from infected or contact lungs were normalized to mock-infected lungs. Brightfield images of lungs were taken using an iPhone 6s and were used to assess the pathology score in a blinded manner. A trained pathologist determined the percentage of lung surface that was affected by lesions, congestion, consolidation, and/or atelectasis using NIH ImageJ (21, 73). Along with the lungs, nasal turbinate was excised and homogenized in PBS using a Precellys tissue homogenizer at 12,000 $\times$ $g$ for 5 min. Supernatants were collected and used to determine viral titers and Nluc activity as described above.

**Statistical analysis.** All data are presented as mean values and SD for each group and were analyzed using Microsoft Excel software. A one-way ANOVA or Student's *t* test was used for statistical

analysis on GraphPad Prism or Microsoft Word software, respectively. Statistical significance was as follows: *, $P < 0.05$; **, $P < 0.01$; ***, $P < 0.001$; ****, $P < 0.0001$; and ns, no significance.

## ACKNOWLEDGMENTS

We are grateful to Thomas Moran at The Icahn School of Medicine at Mount Sinai for providing the SARS-CoV cross-reactive 1C7C7 N protein MAb.

Research on SARS-CoV-2 in the L.M.-S. laboratory was partially supported by grants W81XWH2110103 and W81XWH2110095 from the Department of Defense (DoD) Peer Reviewed Medical Research Program (PRMRP); 1R43AI165089-01, 1R01AI161363-01 and 1R01AI161175-01A1 from the National Institutes of Health (NIH); the Center for Research on Influenza Pathogenesis (CRIP), one of the National Institute of Allergy and Infectious Diseases (NIAID)-funded Centers of Excellence for Influenza Research and Response (CEIRR; contract no. 75N93021C00014); the San Antonio Partnership for Precision Therapeutics; and the San Antonio Medical Foundation. Research in the L.M.-S. laboratory was also partially supported by NIH R01AI145332, R01AI142985, and R01AI141607 and by the DoD W81XWH1910496.

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
