## [Reviewer comments · Microbiology Spectrum]

Microbiology Spectrum

Monitoring SARS-CoV-2 infection using a double reporter-expressing virus

Kevin Chiem, Jun-Gyu Park, Desarey Morales Vasquez, Richard Plemper, Jordi Torrelles, James Kobie, Mark Walter, Chengjin Ye, and Luis Martínez-Sobrido

Corresponding Author(s): Luis Martínez-Sobrido, Texas Biomedical Research Institute

Review Timeline:

Submission Date:	June 23, 2022
Editorial Decision:	July 20, 2022
Revision Received:	July 26, 2022
Accepted:	August 2, 2022

Editor: Daniela Rajao

Reviewer(s): Disclosure of reviewer identity is with reference to reviewer comments included in decision letter(s). The following individuals involved in review of your submission have agreed to reveal their identity: Ghulam Abbas (Reviewer #2)

Transaction Report:

DOI: <https://doi.org/10.1128/spectrum.02379-22>

July 20, 2022

Dr. Luis Martínez-Sobrido
Texas Biomedical Research Institute
San Antonio, TX

Re: Spectrum02379-22 (Monitoring SARS-CoV-2 infection using a double reporter-expressing virus)

Dear Dr. Luis Martínez-Sobrido:

Link Not Available

Sincerely,

Daniela Rajao

Journals Department
Reviewer comments:

Reviewer #1 (Comments for the Author):

This is a game-changing study that reports a carefully constructed and analyzed an advanced double reporter system for SARS-CoV-2. Overall, the authors should be commended for the amount of effort spent on this high quality study. This system could greatly assist preclinical study of COVID-19 infection. A couple of minor comments are presented to consider:

- 1) The high uptake of the virus in the brain by NLuc warrants discussion. Has that been observed in prior literature? Also this was not detected by viral plaque assay.
- 2) SARS-CoV-2 vaccines are mentioned in the introduction but not cited in ref 9-18. Some recent literature reviews could be included (e.g. Rudan et al., doi:10.1097/MCP.0000000000000868, 180, 2022; Mabrouk et al., 10.1002/adma.202107781,2107781, 2022)
- 3) As the NLuc kinetics demonstrate in Fig2, expression is present from 24 to at least 96 hr. Would cells be expected to start dying and losing expression?

4) In the conclusion or discussion, it would be useful for the authors to briefly mention examples of how fluorescence could be used for, what NLuc could be used for, and also how these modalities are expected to track with actual viral load.

Staff Comments:

Preparing Revision Guidelines

Please return the manuscript within 60 days; if you cannot complete the modification within this time period, please contact me. If you do not wish to modify the manuscript and prefer to submit it to another journal, please notify me of your decision immediately so that the manuscript may be formally withdrawn from consideration by Microbiology Spectrum.

July 24th, 2022

Daniela Rajao
Editor, Microbiology Spectrum
Journals Department
Dear Dr. Daniela Rajao,

Thank you very much for your positive reply to our manuscript Spectrum02379-22
“Monitoring SARS-CoV-2 infection using a double reporter-expressing virus” by
Chiem et al.

We are delighted that the reviewer was very positive and interested in our paper. We truly appreciate the reviewer for her/his time and for the constructive comments and suggestions that helped us to improve the manuscript. We have introduced some changes highlighted in gray in the revised version of the document. The following below is a point-by-point response to the comments made by the reviewer.

Reviewer general comments: This is a game-changing study that reports a carefully constructed and analyzed an advanced double reporter system for SARS-CoV-2.

Overall, the authors should be commended for the amount of effort spent on this high quality study. This system could greatly assist preclinical study of COVID-19 infection.

Response: *We appreciate all the positive comments made by this reviewer, indicating that our manuscript is “a game-changing study that reports a carefully constructed and analyzed an advanced double reporter system for SARS-CoV-2”. We also thank the reviewer for indicating that “the authors should be commended for the amount of effort spent on this high quality study”. We also agree with the reviewer that the virus described in our manuscript could assist with preclinical studies of COVID-19 infection.*

Specific comments:

Comment 1: The high uptake of the virus in the brain by NLuc warrants discussion. Has that been observed in prior literature? Also this was not detected by viral plaque assay.

Response: *We concur with the comment made by the reviewer on the importance of the virus detected in the brain using IVIS of the entire mice by assessing Nluc expression. We have previously observed similar levels of Nluc expression in the brain of infected animals starting, in some cases, at day 4 post-infection with a peak of Nluc expression by day 6 post-infection, just before the infected animals succumb to SARS-CoV-2 infection. We have described these findings in our previous manuscript where we described our single reporter expressing rSARS-CoV-2/Nluc (**Analysis of SARS-CoV-2 infection dynamic in vivo using reporter-expressing viruses**. PMID: 34561300). As shown in Figure 7B of the current manuscript, we were able to detect the presence of the virus in two of the animals infected with rSARS-CoV-2/mCherry-Nluc at day 6 post-infection. Notably, this correlated with the levels of Nluc expression detected in the brain of infected mice by luciferase assay (Figure 7A). More importantly, these results correlate with those we previously described with the single reporter expressing rSARS-CoV-2/Nluc (**Analysis of SARS-CoV-2 infection dynamic in vivo using reporter-expressing viruses**. PMID: 34561300). Furthermore, we and others have also demonstrate the presence of wild-type SARS-CoV-2 in the brain of infected K18 hACE2 mice at similar days post-infection (**Lethality of SARS-CoV-2 infection in K18 human angiotensin-converting enzyme 2 transgenic mice**. PMID: 33257679; **Animal Models of COVID-19: Transgenic Mouse Model**. PMID: 35554912).*

Comment 2: SARS-CoV-2 vaccines are mentioned in the introduction but not cited in ref 9-18. Some recent literature reviews could be included (e.g. Rudan et al., doi:10.1097/MCP.0000000000000868, 180, 2022; Mabrouk et al., 10.1002/adma.202107781,2107781, 2022).

Response: *We agree with the comment made by the reviewer and apologize for not including this important references that have now been included in the revised manuscript.*

Comment 3: As the NLuc kinetics demonstrate in Fig2, expression is present from 24 to at least 96 hr. Would cells be expected to start dying and losing expression?

Response: *We thank the reviewer for bringing this important comment to our consideration. We have observed Nluc expression in cells infected with rSARS-CoV-2/Nluc or rSARS-CoV-2/mCherry-Nluc as early as 12 hours post-infection that increase in a time dependent matter and peak at 72 hours (Figure 2B). The decrease in Nluc expression at 96 hours, as indicated by the reviewer, is because of the cytopathic effect induced by the virus that results in losing expression. Similar findings were observed in the manuscript describing the single reporter expressing rSARS-CoV-2/Nluc (**Analysis of SARS-CoV-2 infection dynamic in vivo using reporter-expressing viruses.** PMID: 34561300).*

Comment 4: In the conclusion or discussion, it would be useful for the authors to briefly mention examples of how fluorescence could be used for, what NLuc could be used for, and also how these modalities are expected to track with actual viral load.

Response: *We agree with the comment made by the reviewer and we thank her/him for this suggestion. Following the comment made by the reviewer, we have included in the revised manuscript a brief description on how the fluorescent (mCherry) and luciferase (Nluc) expression from our rSARS-CoV-2/mCherry-Nluc could be used.*

We hope that by taking into account the comments and suggestions made by the reviewer the manuscript has improved, and we hope that now it could be accepted for publication at Microbiology Spectrum. Finally, we want to thank again the reviewer for her/his helpful and constructive comments that contributed to improve our manuscript.

Sincerely,

Luis Martinez-Sobrido, PhD

Texas Biomedical Research Institute.

August 2, 2022

Dr. Luis Martínez-Sobrido
Texas Biomedical Research Institute
San Antonio, TX

Re: Spectrum02379-22R1 (Monitoring SARS-CoV-2 infection using a double reporter-expressing virus)

Dear Dr. Luis Martínez-Sobrido:

Your manuscript has been accepted, and I am forwarding it to the ASM Journals Department for publication. You will be notified when your proofs are ready to be viewed.

Sincerely,

Daniela Rajao
Editor, Microbiology Spectrum

Journals Department
**Monitoring SARS-CoV-2 infection using a double reporter-expressing virus**

Kevin Chiem¹, Jun-Gyu Park¹, Desarey Morales Vasquez¹, Richard K. Plemper², Jordi

B. Torrelles¹, James J. Kobie³, Mark R. Walter⁴, Chengjin Ye^{1*}, Luis Martinez-Sobrido^{1*}

¹ Texas Biomedical Research Institute, San Antonio, Texas 78227, USA

² Center for Translational Antiviral Research, Institute for Biomedical Sciences, Georgia

State University, Atlanta, GA, USA

³ Department of Medicine, Division of Infectious Diseases, University of Alabama at

Birmingham, Birmingham, Alabama 35294, USA

⁴ Department of Microbiology, University of Alabama at Birmingham, Birmingham,

Alabama 35294, USA

*Lead contact email and correspondence:

cye@txbiomed.org

lmartinez@txbiomed.org

**Running title:** A double reporter-expressing recombinant SARS-CoV-2

[revised manuscript text omitted]

and mCherry expression (**Figure 3A**, middle panel) using a microplate reader at 24 hpi.
As internal control, we conducted the microneutralization assay using immunostaining
of rSARS-CoV-2/WT, as previously described (**Figure 3A**, left panel) (71). We
determined the 50% neutralization concentration (NT₅₀) of 1212C2 hMAb using
sigmoidal dose-response curves. The NT₅₀ of 1212C2 hMAb against rSARS-CoV-
2/mCherry (2.4 ng) and rSARS-CoV-2/mCherry-Nluc (2.7 ng) as determined by
fluorescent mCherry expression were similar to that of rSARS-CoV-2/WT (3 ng) using a
classical immunostaining assay, and those reported with the SARS-CoV-2 WA-1 natural
isolate (46, 74). Moreover, NT₅₀ values of 1212C2 hMAb against rSARS-CoV-2/Nluc or

rSARS-CoV-2/mCherry-Nluc determined by Nluc expression (3.0 and 2.0 ng,
respectively) were also comparable to those of rSARS-CoV-2/WT (3 ng). To determine
whether rSARS-CoV-2/mCherry-Nluc could also be used to assess the effectiveness of
antivirals, we quantified the Nluc activity (**Figure 3B**, right panel) and mCherry
expression (**Figure 3B**, middle panel) in Vero E6 cells infected with the single and
double reporter-expressing rSARS-CoV-2 in the presence of serial 3-fold dilutions of
remdesivir. As before, we also included rSARS-CoV-2/WT infected cells stained with
the MAb against the viral N protein as internal control (**Figure 3B**, left panel). Sigmoidal
dose-response curves were developed from reporter expression values and used to
calculate the 50% effective concentration (EC_{50}). The EC_{50} values of remdesivir against
the indicated viruses were similar to each other, regardless of whether the
microneutralization assay used immunostaining (rSARS-CoV-2/WT, 2 μ M; left panel),
fluorescence (rSARS-CoV-2/mCherry, 1.7 μ M; rSARS-CoV-2/mCherry-Nluc, 1.5 μ M;
middle panel), or luciferase (rSARS-CoV-2/Nluc, 1.4 μ M; rSARS-CoV-2/mCherry-Nluc,
1.5 μ M; right panel) (**Figure 3B**). Overall, these results demonstrate the feasibility of
using the rSARS-CoV-2 expressing both mCherry and Nluc reporter genes to reliably
and quickly evaluate the neutralizing and inhibitory properties of NABs and/or antivirals,
respectively, against SARS-CoV-2 based on mCherry and/or Nluc expression,
respectively.

**Characterization of rSARS-CoV-2/mCherry-Nluc in K18 hACE2 transgenic mice**

Next, we characterized the pathogenicity and ability of rSARS-CoV-2/mCherry-Nluc
to replicate in K18 hACE2 transgenic mice using rSARS-CoV-2 expressing individual
fluorescent and bioluminescent reporter genes (rSARS-CoV-2/mCherry and rSARS-

CoV-2/Nluc, respectively), and rSARS-CoV-2/WT as internal control. One group of mice
was infected with a mixture of rSARS-CoV-2/mCherry and rSARS-CoV-2/Nluc. To
assess pathogenicity, groups of K18 hACE2 transgenic mice (n=4/group) were mock-
infected or infected with 10^5 PFUs of the indicated viruses and changes in body weight
(**Figure 4A**) and survival (**Figure 4B**) were monitored for 12 DPI. All mice infected with
rSARS-CoV-2 rapidly lost body weight and succumbed to viral infection (**Figures 4A**
**and 4B**, respectively). Most importantly, the virulence of rSARS-CoV-2/mCherry-Nluc
was shown to be identical to that of our previously reporter viruses expressing individual
mCherry or Nluc (46), or rSARS-CoV-2/WT (71, 76). These data indicate that
expression of the fusion of mCherry and Nluc from rSARS-CoV-2/mCherry-Nluc does
not result in viral attenuation in K18 hACE2 transgenic mouse model as compared to
rSARS-CoV-2/WT.

**Tracking viral dynamics of rSARS-CoV-2/mCherry-Nluc infection and** 519 **pathogenesis in K18 hACE2 transgenic mice**

Since our rSARS-CoV-2/mCherry-Nluc expresses both fluorescent (mCherry) and
luciferase (Nluc) reporter genes, we sought to demonstrate the advantage of using this
newly double reporter-expressing rSARS-CoV-2/mCherry-Nluc to track viral replication
in live animals. Thus, K18 hACE2 transgenic mice were mock-infected or infected with
10^5 PFU of the indicated rSARS-CoV-2 reporter viruses intranasally and Nluc was
monitored at 1, 2, 4, and 6 DPI (**Figure 5A**). In mice infected with rSARS-CoV-2/Nluc or
rSARS-CoV-2/mCherry-Nluc, or co-infected at the same time with rSARS-CoV-2/Nluc
and rSARS-CoV-2/mCherry, we detected Nluc signal as early as 1 DPI, which
increased over time (**Figure 5A**). Since IVIS was conducted in the same mouse, viral

replication and distribution was followed over time (**Figure 5A**) and bioluminescence
intensity around the chest area of the mice was measured in flux (**Figure 5B**). As
expected, Nluc expression increased over time until mice succumbed to SARS-CoV-2
infection, consistent with previous literature, including ours (45). Notably, and as
expected based on the IVIS (**Figure 5A**), Nluc expression was only readily detected in
K18 hACE2 transgenic mice infected with rSARS-CoV-2/Nluc, rSARS-CoV-2/mCherry-
Nluc, or co-infected with both, rSARS-CoV-2/mCherry and rSARS-CoV-2/Nluc (**Figure**
**5B**). No significant differences in flux were observed between the groups of mice
infected with the Nluc-expressing rSARS-CoV-2 mutants (**Figure 5B**).

As luciferase and fluorescence proteins have different properties and could
potentially reveal different readouts as surrogate indicators of viral infection, we next
determined and compared Nluc and mCherry expression during infection *in vivo*. Thus,
K18 hACE2 transgenic mice (n=4) were mock-infected or infected with rSARS-CoV-
2/WT, rSARS-CoV-2/mCherry, rSARS-CoV-2/Nluc, rSARS-CoV-2/mCherry-Nluc, or co-
infected with rSARS-CoV-2/mCherry and rSARS-CoV-2/Nluc, then on 2 and 4 DPI, Nluc
activity in the entire mouse (**Figures 6A and 6B**) and mCherry expression of whole
lungs (**Figures 6C and 6D**) were determined, including the gross pathology score
(**Figure 6E**). Like our previous results (**Figure 5**), an increase in Nluc expression from 2
to 4 DPI was observed in K18 hACE2 transgenic mice infected with rSARS-CoV-2/Nluc,
rSARS-CoV-2/mCherry-Nluc, or co-infected with rSARS-CoV-2/mCherry and rSARS-
CoV-2/Nluc (**Figures 6A**). These results were further confirmed when we determined
the flux in the *in vivo* imaged mice (**Figures 6B**). After quantifying Nluc expression, the
lungs from mock- and rSARS-CoV-2-infected K18 hACE2 transgenic mice were excised

and imaged in the IVIS to determine and quantify mCherry expression (**Figures 6C and**
**6D**, respectively). We only observed detectable levels of mCherry expression in the
lungs of K18 hACE2 transgenic mice infected with rSARS-CoV-2/mCherry, rSARS-coV-
2/mCherry-Nluc, or co-infected with both rSARS-CoV-2/mCherry and rSARS-coV-
2/Nluc (**Figures 6C and 6D**). Notably, levels of mCherry expression, like those of Nluc
were comparable in the lungs of K18 hACE2 transgenic mice infected with the double
reporter-expressing rSARS-CoV-2/mCherry-Nluc than those infected with the single
rSARS-CoV-2/Nluc, or co-infected with rSARS-CoV-2/mCherry and rSARS-CoV-2/Nluc
(**Figures 6C and 6D**). Correlating with *in vivo* and *ex vivo* imaging of the lungs, gross
lung pathology scores were comparable in all rSARS-CoV-2-infected K18 hACE2
transgenic mice and more significant at 4 DPI (**Figure 6E**).

Both Nluc activity and viral titers peaked at 2 DPI in the nasal turbinate of mice
infected with rSARS-CoV-2/Nluc, rSARS-CoV-2/mCherry-Nluc, or co-infected with
rSARS-CoV-2/mCherry and rSARS-CoV-2/Nluc (**Figures 7A and 7B**, left panels)
However, in the lungs, Nluc activity remained the same at 2 and 4 DPI, while viral titers
decreased at 4 DPI as compared to 2 DPI (**Figures 7A and 7B**, middle panels). Nluc
activity in brain homogenates was only evident in the samples from mice infected with
rSARS-CoV-2/Nluc, rSARS-CoV-2/mCherry-Nluc, or both rSARS-CoV-2/mCherry and
rSARS-CoV-2/Nluc and signals increased in a time dependent matter (**Figure 7A**, right
panel). Consistent with previous studies, we were only able to detect SARS-CoV-2 in
the brain of two of the four mice infected with rSARS-CoV-2/mCherry-Nluc at 4 DPI
(**Figure 7B**, right panel) (45). Altogether, these findings demonstrate the feasibility to

[revised manuscript text omitted]

1234